# Improving GANs Using Optimal Transport

**Tim Salimans**[*]
OpenAI
tim@openai.com

**Han Zhang**[*†]
Rutgers University
han.zhang@cs.rutgers.edu

**Alec Radford**
OpenAI
alec@openai.com

**Dimitris Metaxas**
Rutgers University
dnm@cs.rutgers.edu

## Abstract

We present Optimal Transport GAN (OT-GAN), a variant of generative adversarial nets minimizing a new metric measuring the distance between the generator distribution and the data distribution. This metric, which we call mini-batch energy distance, combines optimal transport in primal form with an energy distance defined in an adversarially learned feature space, resulting in a highly discriminative distance function with unbiased mini-batch gradients. Experimentally we show OT-GAN to be highly stable when trained with large mini-batches, and we present state-of-the-art results on several popular benchmark problems for image generation.

## 1 Introduction

Generative modeling is a major sub-field of Machine Learning that studies the problem of how to learn models that generate images, audio, video, text or other data. Applications of generative models include image compression, generating speech from text, planning in reinforcement learning, semi-supervised and unsupervised representation learning, and many others. Since generative models can be trained on unlabeled data, which is almost endlessly available, they have enormous potential in the development of artificial intelligence.

The central problem in generative modeling is how to train a generative model such that the distribution of its generated data will match the distribution of the training data. Generative adversarial nets (GANs) represent an advance in solving this problem, using a neural network *discriminator* or *critic* to distinguish between generated data and training data. The critic defines a distance between the model distribution and the data distribution which the generative model can optimize to produce data that more closely resembles the training data.

A closely related approach to measuring the distance between the distributions of generated data and training data is provided by optimal transport theory. By framing the problem as optimally transporting one set of data points to another, it represents an alternative method of specifying a metric over probability distributions and provides another objective for training generative models. The dual problem of optimal transport is closely related to GANs, as discussed in the next section. However, the primal formulation of optimal transport has the advantage that it allows for closed form solutions and can thus more easily be used to define tractable training objectives that can be evaluated in practice without making approximations. A complication in using primal form optimal transport is that it may give biased gradients when used with mini-batches (see Bellemare et al., 2017) and may therefore be inconsistent as a technique for statistical estimation.

In this paper we present OT-GAN, a variant of generative adversarial nets incorporating primal form optimal transport into its critic. We derive and justify our model by defining a new metric over probability distributions, which we call *Mini-batch Energy Distance*, combining optimal transport

---

[*]equal contribution
[†]work performed during an internship at OpenAI

in primal form with an energy distance defined in an adversarially learned feature space. This combination results in a highly discriminative metric with unbiased mini-batch gradients.

In Section 2 we provide the preliminaries required to understand our work, and we put our contribution into context by discussing the relevant literature. Section 3 presents our main theoretical contribution: Minibatch energy distance. We apply this new distance metric to the problem of learning generative models in Section 4, and show state-of-the-art results in Section 5. Finally, Section 6 concludes by discussing the strengths and weaknesses of the proposed method, as well as directions for future work.

## 2  GANs AND OPTIMAL TRANSPORT

Generative adversarial nets (Goodfellow et al., 2014) were originally motivated using game theory: A generator $g$ and a discriminator $d$ play a zero-sum game where the generator maps noise $\mathbf{z}$ to simulated images $\mathbf{y} = g(\mathbf{z})$ and where the discriminator tries to distinguish the simulated images $\mathbf{y}$ from images $\mathbf{x}$ drawn from the distribution of training data $p$. The discriminator takes in each image $\mathbf{x}$ and $\mathbf{y}$ and outputs an estimated probability that the given image is real rather than generated. The discriminator is rewarded for putting high probability on the correct classification, and the generator is rewarded for fooling the discriminator. The goal of training is then to find a pair of $(g, d)$ for which this game is at a Nash equilibrium. At such an equilibrium, the generator minimizes its loss, or negative game value, which can be defined as

$$L_g = \sup_d \mathbb{E}_{\mathbf{x} \sim p} \log[d(\mathbf{x})] + \mathbb{E}_{\mathbf{y} \sim g} \log[1 - d(\mathbf{y})] \tag{1}$$

Arjovsky et al. (2017) re-interpret GANs in the framework of optimal transport theory. Specifically, they propose the *Earth-Mover distance* or *Wasserstein-1 distance* as a good objective for generative modeling:

$$D_{\text{EMD}}(p, g) = \inf_{\gamma \in \Pi(p, g)} \mathbb{E}_{\mathbf{x}, \mathbf{y} \sim \gamma} c(\mathbf{x}, \mathbf{y}), \tag{2}$$

where $\Pi(p, g)$ is the set of all joint distributions $\gamma(\mathbf{x}, \mathbf{y})$ with marginals $p(\mathbf{x}), g(\mathbf{y})$, and where $c(\mathbf{x}, \mathbf{y})$ is a cost function that Arjovsky et al. (2017) take to be the Euclidean distance. If the $p(\mathbf{x})$ and $g(\mathbf{y})$ distributions are interpreted as piles of earth, the Earth-Mover distance $D_{\text{EMD}}(p, g)$ can be interpreted as the minimum amount of "mass" that $\gamma$ has to transport to turn the generator distribution $g(\mathbf{y})$ into the data distribution $p(\mathbf{x})$. For the right choice of cost $c$, this quantity is a *metric* in the mathematical sense, meaning that $D_{\text{EMD}}(p, g) \geq 0$ and $D_{\text{EMD}}(p, g) = 0$ if and only if $p = g$. Minimizing the Earth-Mover distance in $g$ is thus a valid method for deriving a statistically consistent estimator of $p$, provided $p$ is in the model class of our generator $g$.

Unfortunately, the minimization over $\gamma$ in Equation 2 is generally intractable, so Arjovsky et al. (2017) turn to the dual formulation of this optimal transport problem:

$$D_{\text{EMD}}(p, g) = \sup_{\|f\|_L \leq 1} \mathbb{E}_{\mathbf{x} \sim p} f(\mathbf{x}) - \mathbb{E}_{\mathbf{y} \sim g} f(\mathbf{y}), \tag{3}$$

where we have replaced the minimization over $\gamma$ with a maximization over the set of 1-Lipschitz functions. This optimization problem is generally still intractable, but Arjovsky et al. (2017) argue that it is well approximated by using the class of neural network GAN discriminators or critics described earlier in place of the class of 1-Lipschitz functions, provided we bound the norm of their gradient with respect to the image input. Making this substitution, the objective becomes quite similar to that of our original GAN formulation in Equation 1.

In followup work Gulrajani et al. (2017) propose a different method of bounding the gradients in the class of allowed critics, and provide strong empirical results supporting this interpretation of GANs. In spite of their success, however, we should note that GANs are still only able to solve this optimal transport problem approximately. The optimization with respect to the critic cannot be performed perfectly, and the class of obtainable critics only very roughly corresponds to the class of 1-Lipschitz functions. The connection between GANs and dual form optimal transport is further explored by Bousquet et al. (2017) and Genevay et al. (2017a), who extend the analysis to different optimal transport costs and to a broader model class including latent variables.

An alternative approach to generative modeling is chosen by Genevay et al. (2017b) who instead chose to approximate the primal formulation of optimal transport. They start by taking an entropically smoothed generalization of the Earth Mover distance, called the *Sinkhorn distance* (Cuturi,

2013):

$$D_{\text{Sinkhorn}}(p, g) = \inf_{\gamma \in \Pi_\beta(p,g)} \mathbb{E}_{\mathbf{x}, \mathbf{y} \sim \gamma}\, c(\mathbf{x}, \mathbf{y}), \tag{4}$$

where the set of allowed joint distribution $\Pi_\beta$ is now restricted to distributions with entropy of at least some constant $\beta$. Genevay et al. (2017b) then approximate this distance by evaluating it on mini-batches of data $\mathbf{X}, \mathbf{Y}$ consisting of $K$ data vectors $\mathbf{x}, \mathbf{y}$. The cost function $c$ then gives rise to a $K \times K$ transport cost matrix $C$, where $C_{i,j} = c(\mathbf{x}_i, \mathbf{y}_j)$ tells us how expensive it is to transport the $i$-th data vector $\mathbf{x}_i$ in mini-batch $\mathbf{X}$ to the $j$-th data vector $\mathbf{y}_j$ in mini-batch $\mathbf{Y}$. Similarly, the coupling distribution $\gamma$ is replaced by a $K \times K$ matrix $M$ of *soft matchings* between these $i, j$ elements, which is restricted to the set of matrices $\mathcal{M}$ with all positive entries, with all rows and columns summing to one, and with sufficient entropy $-\operatorname{Tr}[M\log(M^{\mathrm{T}})] \geq \alpha$. The resulting distance, evaluated on a minibatch, is then

$$\mathcal{W}_c(X, Y) = \inf_{M \in \mathcal{M}} \operatorname{Tr}[MC^{\mathrm{T}}]. \tag{5}$$

In practice, the minimization over the soft matchings $M$ can be found efficiently on the GPU using the Sinkhorn algorithm. Consequently, Genevay et al. (2017b) call their method of using Equation 5 in generative modeling *Sinkhorn AutoDiff*.

The great advantage of this mini-batch Sinkhorn distance is that it is fully tractable, eliminating the instabilities often experienced with GANs due to imperfect optimization of the critic. However, a disadvantage is that the expectation of Equation 5 over mini-batches is no longer a valid metric over probability distributions. Viewed another way, the gradients of Equation 5, for fixed mini-batch size, are not unbiased estimators of the gradients of our original optimal transport problem in Equation 4. For this reason, Bellemare et al. (2017) propose to instead use the *Energy Distance*, also called *Cramer Distance*, as the basis of generative modeling:

$$D_{\text{ED}}(p, g) = \sqrt{2\,\mathbb{E}[\|\mathbf{x} - \mathbf{y}\|] - \mathbb{E}[\|\mathbf{x} - \mathbf{x}'\|] - \mathbb{E}[\|\mathbf{y} - \mathbf{y}'\|]}, \tag{6}$$

where $\mathbf{x}, \mathbf{x}'$ are independent samples from data distribution $p$ and $\mathbf{y}, \mathbf{y}'$ independent samples from the generator dsitribution $g$. In *Cramer GAN* they propose training the generator by minimizing this distance metric, evaluated in a latent space which is learned by the GAN critic.

In the next section we propose a new metric for generative modeling, combining the insights of GANs and optimal transport. Although our work was performed concurrently to that by Genevay et al. (2017b) and Bellemare et al. (2017), it can be understood most easily as forming a synthesis of the ideas used in Sinkhorn AutoDiff and Cramer GAN.

## 3 MINI-BATCH ENERGY DISTANCE

As discussed in the last section, most previous work in generative modeling can be interpreted as minimizing a distance $D(g, p)$ between a *generator distribution* $g(\mathbf{x})$ and the *data distribution* $p(\mathbf{x})$, where the distributions are defined over a single vector $\mathbf{x}$ which we here take to be an image. However, in practice deep learning typically works with *mini-batches* of images $\mathbf{X}$ rather than individual images. For example, a GAN generator is typically implemented as a high dimensional function $G(\mathbf{Z})$ that turns a mini-batch of random noise $\mathbf{Z}$ into a mini-batch of images $\mathbf{X}$, which the GAN discriminator then compares to a mini-batch of images from the training data. The central insight of *Mini-batch GAN* (Salimans et al., 2016) is that it is strictly more powerful to work with the distributions over mini-batches $g(\mathbf{X}), p(\mathbf{X})$ than with the distributions over individual images. Here we further pursue this insight and propose a new distance over mini-batch distributions $D[g(\mathbf{X}), p(\mathbf{X})]$ which we call the *Mini-batch Energy Distance*. This new distance combines optimal transport in primal form with an energy distance defined in an adversarially learned feature space, resulting in a highly discriminative distance function with unbiased mini-batch gradients.

In order to derive our new distance function, we start by generalizing the energy distance given in Equation 6 to general non-Euclidean distance functions $d$. Doing so gives us the *generalized energy distance*:

$$D_{\text{GED}}(p, g) = \sqrt{2\,\mathbb{E}[d(\mathbf{X}, \mathbf{Y})] - \mathbb{E}[d(\mathbf{X}, \mathbf{X}')] - \mathbb{E}[d(\mathbf{Y}, \mathbf{Y}')]}, \tag{7}$$

where $\mathbf{X}, \mathbf{X}'$ are independent samples from distribution $p$ and $\mathbf{Y}, \mathbf{Y}'$ independent samples from $g$. This distance is typically defined for individual samples, but it is valid for general random objects, including mini-batches like we assume here. The energy distance $D_{\text{GED}}(p, g)$ is a metric, in the

mathematical sense, as long as the distance function $d$ is a metric (Klebanov et al., 2005). Under this condition, meaning that $d$ satisfies the triangle inequality and several other conditions, we have that $D(p, g) \geq 0$, and $D(p, g) = 0$ if and only if $p = g$.

Using individual samples $\mathbf{x}, \mathbf{y}$ instead of minibatches $\mathbf{X}, \mathbf{Y}$, Sejdinovic et al. (2013) showed that such generalizations of the energy distance can equivalently be viewed as a form of *maximum mean discrepancy*, where the MMD kernel $k$ is related to the distance function $d$ by $d(\mathbf{x}, \mathbf{x}') \equiv k(\mathbf{x}, \mathbf{x}) + k(\mathbf{x}', \mathbf{x}') - 2k(\mathbf{x}, \mathbf{x}')$. We find the energy distance perspective more intuitive here and follow Cramer GAN in using this perspective instead.

We are free to choose any metric $d$ for use in Equation 7, but not all choices will be equally discriminative when used for generative modeling. Here, we choose $d$ to be the entropy-regularized Wasserstein distance, or Sinkhorn distance, as defined for mini-batches in Equation 5. Although the average over mini-batch Sinkhorn distances is not a valid metric over probability distributions $p, g$, resulting in the biased gradients problem discussed in Section 2, the Sinkhorn distance is a valid metric between *individual mini-batches*, which is all we require for use inside the generalized energy distance.

Putting everything together, we arrive at our final distance function over distributions, which we call the *Minibatch Energy Distance*. Like with the Cramer distance, we typically work with the squared distance, which we define as

$$D^2_{\mathrm{MED}}(p, g) = 2 \, \mathbb{E}[\mathcal{W}_c(\mathbf{X}, \mathbf{Y})] - \mathbb{E}[\mathcal{W}_c(\mathbf{X}, \mathbf{X}')] - \mathbb{E}[\mathcal{W}_c(\mathbf{Y}, \mathbf{Y}')], \tag{8}$$

where $\mathbf{X}, \mathbf{X}'$ are independently sampled mini-batches from distribution $p$ and $\mathbf{Y}, \mathbf{Y}'$ are independent mini-batches from $g$. We include the subscript $c$ to make explicit that this distance depends on the choice of transport cost function $c$, which we will learn adversarially as discussed in Section 4.

In comparison with the original Sinkhorn distance (Equation 5 the loss function for training $g$ implied by this metric adds a repulsive term $- \mathcal{W}_c(\mathbf{Y}, \mathbf{Y}')$ to the attractive term $\mathcal{W}_c(\mathbf{X}, \mathbf{Y})$. Like with the energy distance used by Cramer GAN, this is what makes the resulting mini-batch gradients unbiased and the objective statistically consistent. However, unlike the plain energy distance, the mini-batch energy distance $D^2_{\mathrm{MED}}(p, g)$ still incorporates the primal form optimal transport of the Sinkhorn distance, which in Section 5 we show leads to much stronger discriminative power and more stable generative modeling.

In concurrent work, Genevay et al. (2018) independently propose a very similar loss function to (8), but using a single sample from the data and generator distributions. We obtained best results using two independently sampled minibatches from each distribution.

## 4 OPTIMAL TRANSPORT GAN (OT-GAN)

In the last section we defined the mini-batch energy distance which we propose using for training generative models. However, we left undefined the transport cost function $c(\mathbf{x}, \mathbf{y})$ on which it depends. One possibility would be to choose $c$ to be some fixed function over vectors, like Euclidean distance, but we found this to perform poorly in preliminary experiments. Although minimizing the mini-batch energy distance $D^2_{MED}(p, g)$ guarantees *statistical consistency* for simple fixed cost functions $c$ like Euclidean distance, the resulting *statistical efficiency* is generally poor in high dimensions. This means that there typically exist many bad distributions distributions $g$ for which $D^2_{MED}(p, g)$ is so close to zero that we cannot tell $p$ and $g$ apart without requiring an enormous sample size. To solve this we propose learning the cost function adversarially, so that it can adapt to the generator distribution $g$ and thereby become more discriminative. In practice we implement this by defining $c$ to be the cosine distance between vectors $v_\eta(\mathbf{x})$ and $v_\eta(\mathbf{y})$, where $v_\eta$ is a deep neural network that maps the images in our mini-batch into a learned latent space. That is we define the transport cost to be

$$c_\eta(\mathbf{x}, \mathbf{y}) = 1 - \frac{v_\eta(\mathbf{x}) \cdot v_\eta(\mathbf{y})}{\|v_\eta(\mathbf{x})\|_2 \|v_\eta(\mathbf{y})\|_2},$$

where we choose $\eta$ to maximize the resulting minibatch energy distance.

In practice, training our generative model $g_\theta$ and our adversarial transport cost $c_\eta$ is done by alternating gradient descent as is standard practice in GANs (Goodfellow et al., 2014). Here we choose

to update the generator more often than we update our critic. This is contrary to standard practice (e.g. Arjovsky et al., 2017) and ensures our cost function $c$ does not become degenerate. If $c$ were to assign zero transport cost to two non-identical regions in image space, the generator would quickly adjust to take advantage of this. Similar to how a quickly adapting critic controls the generator in standard GANs, this works the other way around in our case. Contrary to standard GANs, our generator has a well defined and statistically consistent training objective even when the critic is not updated, as long as the cost function $c$ is not degenerate. We also investigated forcing $v_\eta$ to be one-to-one by parameterizing it using a RevNet Gomez et al. (2017), thereby ensuring $c$ cannot degenerate, but this proved unnecessary if the generator is updated often enough.

Our full training procedure is described in Algorithm 1, and is visually depicted in Figure 1. Here we compute the matching matrix $M$ in $\mathcal{W}_c$ using the Sinkhorn algorithm. Unlike Genevay et al. (2017b) we do not backpropagate through this algorithm. Ignoring the gradient flow through the matchings $M$ is justified by the envelope theorem (see e.g. Carter, 2001): Since $M$ is chosen to minimize $\mathcal{W}_c$, the gradient of $\mathcal{W}_c$ with respect to this variable is zero (when projected into the allowed space $\mathcal{M}$). Algorithm 1 assumes we use standard SGD for optimization, but we are free to use other optimizers. In our experiments we use Adam (Kingma & Ba, 2014).

Our algorithm for training generative models can be generalized to include conditional generation of images given some side information $s$, such as a text-description of the image or a label. When generating an image $\mathbf{y}$ we simply draw $s$ from the training data and condition the generator on it. The rest of the algorithm is identical to Algorithm 1 but with $(\mathbf{Y}, S)$ in place of $\mathbf{Y}$, and similar substitutions for $\mathbf{X}, \mathbf{X}', \mathbf{Y}'$. The full algorithm for conditional generation is detailed in Algorithm 2 in the appendix.

---

**Algorithm 1** Optimal Transport GAN (OT-GAN) training algorithm with step size $\alpha$, using mini-batch SGD for simplicity

---

**Require:** $n_{gen}$, the number of iterations of the generator per critic iteration
**Require:** $\eta_0$, initial critic parameters. $\theta_0$, initial generator parameters
 1: **for** $t = 1$ to $N$ **do**
 2:     Sample $\mathbf{X}, \mathbf{X}'$ two independent mini-batches from real data, and $\mathbf{Y}, \mathbf{Y}'$ two independent mini-batches from the generated samples
 3:     $\mathcal{L} = \mathcal{W}_c(\mathbf{X}, \mathbf{Y}) + \mathcal{W}_c(\mathbf{X}, \mathbf{Y}') + \mathcal{W}_c(\mathbf{X}', \mathbf{Y}) + \mathcal{W}_c(\mathbf{X}', \mathbf{Y}') - 2\,\mathcal{W}_c(\mathbf{X}, \mathbf{X}') - 2\,\mathcal{W}_c(\mathbf{Y}, \mathbf{Y}')$
 4:     **if** $t \bmod n_{gen} + 1 = 0$ **then**
 5:         $\eta \leftarrow \eta + \alpha \cdot \nabla_\eta \mathcal{L}$
 6:     **else**
 7:         $\theta \leftarrow \theta - \alpha \cdot \nabla_\theta \mathcal{L}$
 8:     **end if**
 9: **end for**

---

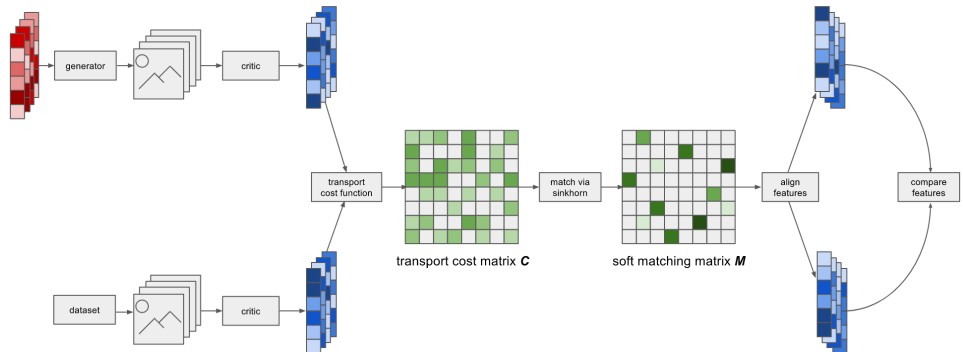

Figure 1: Illustration of OT-GAN. Mini-batches from the generator and training data are embedded into a learned feature space via the critic. A transport cost matrix is calculated between the two mini-batches of features. Soft matching aligns features across mini-batches and aligned features are compared. The figure only illustrates the distance calculation between one pair of mini-batches whereas several are computed.

## 5 EXPERIMENTS

In this section, we demonstrate the improved stability and consistency of the proposed method on five different datasets with increasing complexity.

### 5.1 MIXTURE OF GAUSSIAN DATASET

One advantage of OT-GAN compared to regular GAN is that for any setting of the transport cost $c$, i.e. any fixed critic, the objective is statistically consistent for training the generator $g$. Even if we stop updating the critic, the generator should thus never diverge. With a bad fixed cost function $c$ the signal for learning $g$ may be very weak, but at least it should never point in the wrong direction. We investigate whether this theoretical property holds in practice by examining a simple toy example. We train generative models using different types of GAN on a 2D mixture of 8 Gaussians, with means arranged on a circle. The goal for the generator is to recover all 8 modes. For the proposed method and all the baseline methods, the architectures are simple MLPs with ReLU activations. A similar experimental setting has been considered in (Metz et al., 2017; Li et al., 2017) to demonstrate the mode coverage behavior of various GAN models. There, GANs using mini-batch features, DAN-S (Li et al., 2017), are shown to capture all the 8 modes when training converges. To test the consistency of GAN models, we stop updating the discriminator after 15k iterations and visualize the generator distribution for an additional 25K iterations. As shown in Figure 2, mode collapse occurs in a mini-batch feature GAN after a few thousand iterations training with a fixed discriminator. However, using the mini-batch energy distance, the generator does not diverge and the generated samples still cover all 8 modes of the data.

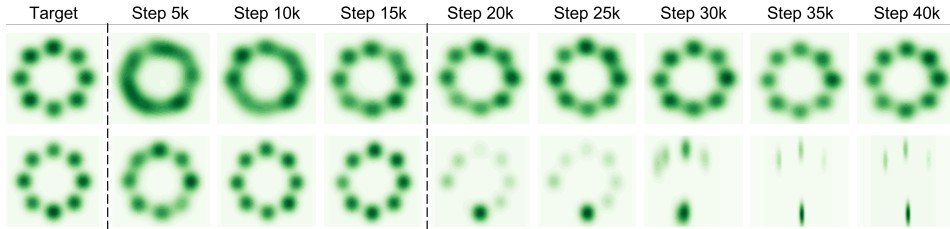

Figure 2: Results for consistency when fixing the critic on data generated from 8 Gaussian mixtures. The first column shows the data distribution. The top row shows the training results of OT-GAN using mini-batch energy distance. The bottom row shows the training result with the original GAN loss (DAN-S). The latter collapses to 3 out of 8 modes after fixing the discriminator, while OT-GAN remains consistent.

### 5.2 CIFAR-10

CIFAR-10 is a well-studied dataset of $32\times32$ color images for generative models (Krizhevsky, 2009). We use this data set to investigate the importance of the different design decisions made with OT-GAN, and we compare the visual quality of its generated samples with other state-of-the-art GAN models. Our model and the other reported results are trained in an unsupervised manner. We choose "inception score" (Salimans et al., 2016) as numerical assessment to compare the visual quality of samples generated by different models. Our generator and critic are standard convnets, similar to those used by DCGAN (Radford et al., 2015), but without any batch normalization, layer normalization, or other stabilizing additions. Appendix B contains additional architecture and training details.

We first investigate the effect of batch size on training stability and sample quality. As shown in Figure 3, training is not very stable when the batch size is small (i.e. 200). As batch size increases, training becomes more stable and the inception score of samples increases. Unlike previous methods, our objective (the minibatch energy distance, Section 3) depends on the chosen minibatch size: Larger minibatches are more likely to cover many modes of the data distribution, thereby not only yielding lower variance estimates but also making our distance metric more discriminative. To reach the large batch sizes needed for optimal performance we make use of multi GPU training. In this work we only use up to 8 GPUs per experiment, but we anticipate more GPUs to be useful when using larger models.

In Figure 4 we present the samples generated by our model trained with a batch size of 8000. In addition, we also compare with the sample quality of other state-of-the-art GAN models in Table 1. OT-GAN achieves a score of 8.47 ± .12, outperforming all baseline models.

To evaluate the importance of using optimal transport in OT-GAN, we repeat our CIFAR-10 experiment with *random* matching of samples. Our minibatch energy distance objective remains valid when we match samples randomly rather than using optimal transport. In this case the minibatch energy distance reduces to the regular (generalized) energy distance. We repeat our CIFAR-10 experiment and train a generator with the same architecture and hyperparameters as above, but with random matching of samples instead of optimal transport. The highest resulting Inception score achieved during the training process is 4.64 using this approach, as compared to 8.47 with optimal transport. Figure 5 shows a random sample from the resulting model.

| Method | Inception score |
|---|---|
| Real Data | 11.95 ± .12 |
| DCGAN | 6.16 ± .07 |
| Improved GAN | 6.86 ± .06 |
| Denoising FM | 7.72 ± .13 |
| WGAN-GP | 7.86 ± .07 |
| **OT-GAN** | **8.47 ± .12** |

Table 1: Inception scores on CIFAR-10. All the models are trained in an unsupervised manner.

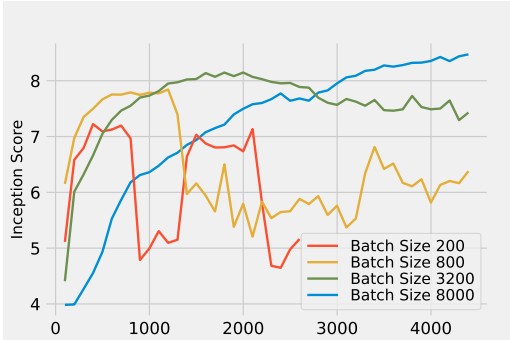

Figure 3: CIFAR-10 inception score over the course of training for different batch sizes.

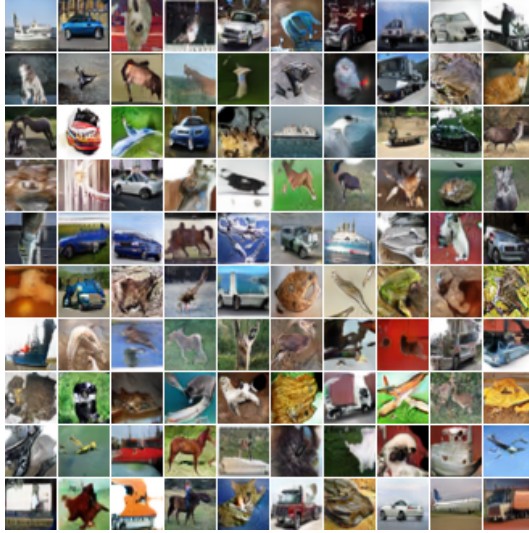

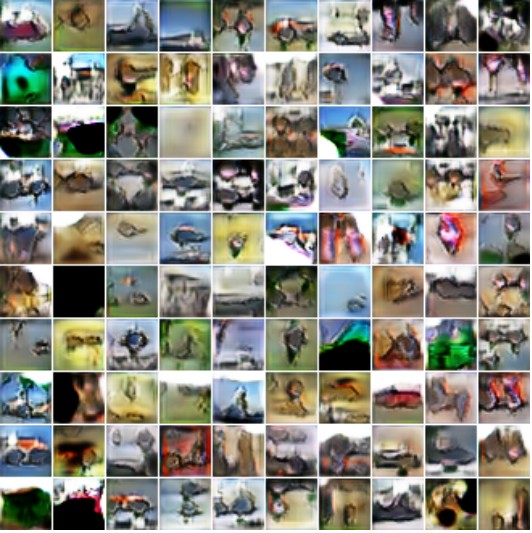

Figure 4: Samples generated by OT-GAN on CIFAR-10, without using labels.

Figure 5: Samples generated without using optimal transport.

## 5.3 IMAGENET DOGS

To illustrate the ability of OT-GAN in generating high quality images on more complex data sets, we train OT-GAN to generate 128×128 images on the dog subset of ImageNet (Russakovsky et al., 2015). A smaller batch size of 2048 is used due to GPU memory contraints. As shown in Figure 6, the samples generated by OT-GAN contain less nonsensical images, and the sample quality is significantly better than that of a tuned DCGAN variant which still suffers from mode collapse. The superior image quality is confirmed by the inception score achieved by OT-GAN(8.97±0.09) on this dataset, which outperforms that of DCGAN(8.19±0.11)

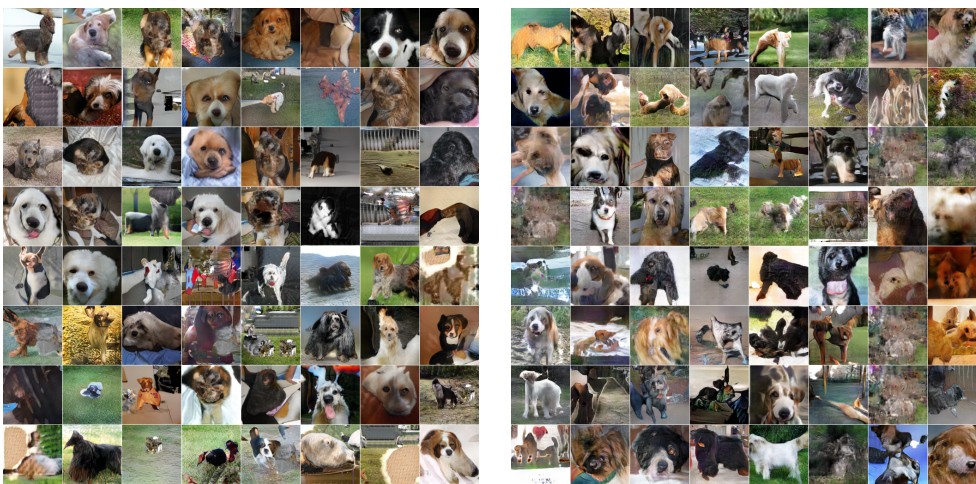

Figure 6: ImageNet Dog subset samples generated by OT-GAN (left) and DCGAN (right).

## 5.4 CONDITIONAL GENERATION OF BIRDS

To further demonstrate the effectiveness of the proposed method on conditional image synthesis, we compare OT-GAN with state-of-the-art models on text-to-image generation (Reed et al., 2016b;a; Zhang et al., 2017). As shown in Table 2, the images generated by OT-GAN with batch size 2048 also achieve the best inception score here. Example images generated by our conditional generative model on the CUB test set are presented in Figure 7.

| Method | GAN-INT-CLS | GAWWN | StackGAN | OT-GAN |
|---|---|---|---|---|
| Inception Score | 2.88 ± .04 | 3.62 ± .07 | 3.70 ± .04 | **3.84 ± .05** |

Table 2: Inception scores by state-of-the-art methods (Reed et al., 2016b;a; Zhang et al., 2017) and the proposed OT-GAN on the CUB test set. Higher inception scores mean better image quality.

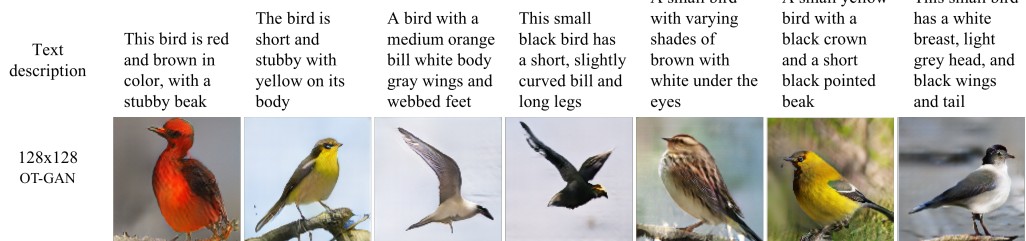

Figure 7: Bird example images generated by conditional OT-GAN

## 6 DISCUSSION

We have presented OT-GAN, a new variant of GANs where the generator is trained to minimize a novel distance metric over probability distributions. This metric, which we call mini-batch energy distance, combines optimal transport in primal form with an energy distance defined in an adversarially learned feature space, resulting in a highly discriminative distance function with unbiased mini-batch gradients. OT-GAN was shown to be uniquely stable when trained with large mini-batches and to achieve state-of-the-art results on several common benchmarks.

One downside of OT-GAN, as currently proposed, is that it requires large amounts of computation and memory. We achieve the best results when using very large mini-batches, which increases the time required for each update of the parameters. All experiments in this paper, except for the mixture of Gaussians toy example, were performed using 8 GPUs and trained for several days. In future work we hope to make the method more computationally efficient, as well as to scale up our approach to multi-machine training to enable generation of even more challenging and high resolution image data sets.

A unique property of OT-GAN is that the mini-batch energy distance remains a valid training objective even when we stop training the critic. Our implementation of OT-GAN updates the generative model more often than the critic, where GANs typically do this the other way around (see e.g. Gulrajani et al., 2017). As a result we learn a relatively stable transport cost function $c(\mathbf{x}, \mathbf{y})$, describing how (dis)similar two images are, as well as an image embedding function $v_\eta(\mathbf{x})$ capturing the geometry of the training data. Preliminary experiments suggest these learned functions can be used successfully for unsupervised learning and other applications, which we plan to investigate further in future work.

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

## A    CONDITIONAL GENERATION

---

**Algorithm 2** Conditional Optimal Transport GAN (OT-GAN) training algorithm with step size $\alpha$, using minibatch SGD for simplicity

---

**Require:** $n_{gen}$, the number of iterations of the generator per critic iteration
**Require:** $\eta_0$, initial critic parameters. $\theta_0$, initial generator parameters
  1: **for** $t = 1$ to $N$ **do**
  2:     Sample $(\mathbf{X}, S), (\mathbf{X}', S')$ two independent mini-batches from real data, with side information, and $(\mathbf{Y}, S), (\mathbf{Y}', S')$ two independent mini-batches from the generator, re-using the same side information
  3:     $\mathcal{L} = \mathcal{W}_c[(\mathbf{X}, S), (\mathbf{Y}', S')] + \mathcal{W}_c[(\mathbf{X}', S'), (\mathbf{Y}, S)] - \mathcal{W}_c[(\mathbf{X}, S), (\mathbf{X}', S')] - \mathcal{W}_c[(\mathbf{Y}, S), (\mathbf{Y}', S')]$
  4:     **if** $t \bmod n_{gen} + 1 = 0$ **then**
  5:         $\eta \leftarrow \eta + \alpha \cdot \nabla_\eta \mathcal{L}$
  6:     **else**
  7:         $\theta \leftarrow \theta - \alpha \cdot \nabla_\theta \mathcal{L}$
  8:     **end if**
  9: **end for**

---

## B    CIFAR-10 ARCHITECTURE AND TRAINING DETAILS

The generator and critic are implemented as convolutional networks. Their architectures are loosely based on DCGAN with various modifications. Weight normalization and data-dependent initialization (Salimans & Kingma, 2016) are used for both. The generator maps latent codes sampled from a 100 dimensional uniform distribution between -1 and 1 to $32 \times 32$ color images. The main module of the generator is a 2x2 nearest-neighbor upsampling operation followed by a convolution with a $5 \times 5$ kernel using gated linear units (Dauphin et al., 2016). The main module of the critic is a convolution with a $5 \times 5$ kernel and stride 2 using the concatenated ReLU activation function (Shang et al., 2016). Notably, the generator and critic do not use an activation normalization technique such as batch or layer normalization. We train the model using Adam with a learning rate of $3 \times 10^{-4}$, $\beta_1 = 0.5$, $\beta_2 = 0.999$. We update the generator 3 times for every critic update. OT-GAN includes two additional hyperparameters for the Sinkhorn algorithm, the number of iterations to run the algorithm and $\frac{1}{\lambda}$ which is the entropy penalty of alignments. Initial tuning found a value of 500 to work well for both.

| operation | activation | kernel | stride | output shape |
|---|---|---|---|---|
| $z$ | | | | 100 |
| linear | GLU | | | 16384 |
| reshape | | | | $1024 \times 4 \times 4$ |
| 2x NN upsample | | | | $1024 \times 8 \times 8$ |
| convolution | GLU | $5 \times 5$ | 1 | $512 \times 8 \times 8$ |
| 2x NN upsample | | | | $512 \times 16 \times 16$ |
| convolution | GLU | $5 \times 5$ | 1 | $256 \times 16 \times 16$ |
| 2x NN upsample | | | | $256 \times 32 \times 32$ |
| convolution | GLU | $5 \times 5$ | 1 | $128 \times 32 \times 32$ |
| convolution | tanh | $5 \times 5$ | 1 | $3 \times 32 \times 32$ |

Table 3: Generator architecture for CIFAR-10.

| operation | activation | kernel | stride | output shape |
|---|---|---|---|---|
| convolution | CReLU | $5 \times 5$ | 1 | $256 \times 32 \times 32$ |
| convolution | CReLU | $5 \times 5$ | 2 | $512 \times 16 \times 16$ |
| convolution | CReLU | $5 \times 5$ | 2 | $1024 \times 8 \times 8$ |
| convolution | CReLU | $5 \times 5$ | 2 | $2048 \times 4 \times 4$ |
| reshape | | | | 32768 |
| l2 normalize | | | | 32768 |

Table 4: Critic architecture for CIFAR-10.

## C ADVERSARIALLY LEARNING THE TRANSPORT COST FUNCTION

To illustrate the importance of learning the transport cost function adversarially, we repeat our CIFAR-10 experiment using cosine distance defined in the original feature space:

$$c(\mathbf{x}, \mathbf{y}) = 1 - \frac{\mathbf{x} \cdot \mathbf{y}}{\| \mathbf{x} \|_2 \| \mathbf{y} \|_2},$$

where $\mathbf{x}, \mathbf{y}$ are original image pixel values. In this case, only the transport cost function is a fixed distance function, but all the rest experiment settings are the same as those of OT-GAN. The highest inception score during the training process is 4.93, as compared to 8.47 when learning cost function adversarially using another neural network. The generated samples are shown in Figure 8.

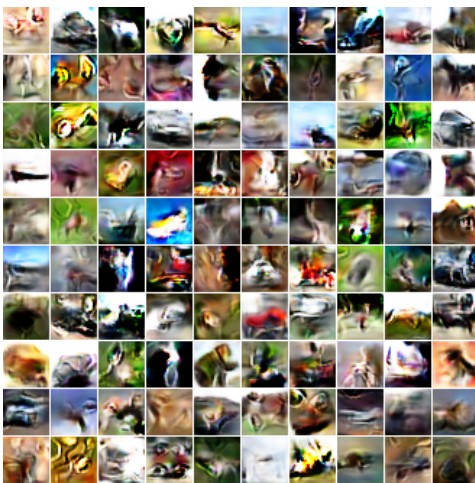

Figure 8: CIFAR-10 Samples generated without adversarially learning the cost function.

## D  MODEL COLLAPSE AND SAMPLE DIVERSITY

To further investigate sample diversity and mode collapse in GANs, we train the same generator using DCGAN and OT-GAN on the Imagenet dog data set for a large number of epochs. For DCGAN we observe mode collapse starting to occur after about 900 epochs, as indicated in figure 9. The model does not recover from this if we continue training. We have observed similar behavior for many other types of GAN. For OT-GAN we continued to train for 13000 epochs on this data set but never observed any mode collapse or reduction in sample diversity.

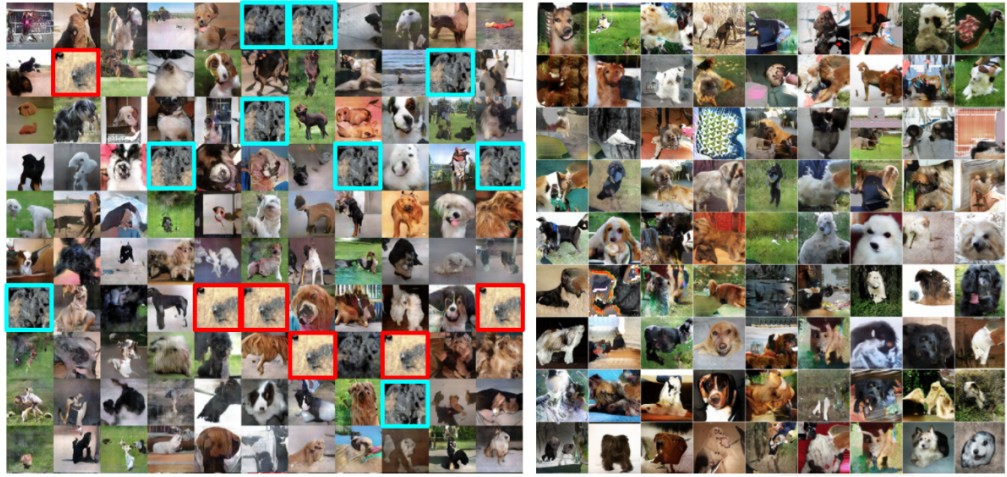

Figure 9: Imagenet dog samples generated with DCGAN (left) after 900 epochs and OT-GAN (right) after 13000 epochs. When training long enough, DCGAN suffers from mode collapse as indicated by the highlighted samples. We did not observe any mode collapse for OT-GAN, even when training for many more epochs.

