# OpenReview forum: "Improving GANs Using Optimal Transport"
_ICLR.cc/2018/Conference — Accept (Poster)_

### Official Review · AnonReviewer1 · 2017-11-25
**New algorithm motivated theoretically, good results, honest writing**

**Rating:** 8
**Confidence:** 4

**Review:**

The paper introduces a new algorithm for training GANs based on the Earth Mover’s distance. In order to avoid biased gradients, the authors use the dual form of the distance on mini-batches, to make it more robust. To compute the distance between mini batches, they use the Sinkhorn distance. Unlike the original Sinkhorn distance paper, they use the dual form of the distance and do not have biased gradients. Unlike the Cramer GAN formulation, they use a mini-batch distance allowing for a better leverage of the two distributions, and potentially decrease variance in gradients.

Evaluation: the paper shows good results a battery of tasks, including a standard toy example, CIFAR-10 and conditional image generation, where they obtain better results than StackGAN.

The paper is honest about its shortcomings, in the current set up the model requires a lot of computation, with best results obtained using a high batch size.

Would like to see:
  * a numerical comparison with Cramer GAN, to see whether the additional  computational cost is worth the gains.
  * Cramer GAN shows an increase in diversity, would like to see an analog experiment for conditional generation, like figure 3 in the Cramer GAN paper.

---

> ### Author Response · Authors · 2018-01-02
> **Happy new year and thank you for your review!**
>
> Regarding your two requests:
>
> * The Cramer GAN paper did not report inception scores on the usual data sets, and they do not provide code, so numerical comparison to their work is a bit difficult. I searched for public implementations of Cramer GAN, and the best one I could find is this one: https://github.com/pfnet-research/chainer-gan-lib According to the inception scores reported here, our method performs much better than Cramer GAN.
>
> * We are in correspondence with the authors of Cramer GAN about the details of this conditional generation experiment. So far we have not yet been able to replicate their exact setup. We hope to be able to include this experiment in the paper soon. Related to what you’re asking for, we have added an additional experiment to the paper investigating sample diversity and mode collapse in OT-GAN as compared to DCGAN (appendix D). What we find here is that DCGAN (as well as other variants of GAN) still shows mode collapse when training for longer periods of time. For OT-GAN we see no mode collapse, even when we keep training for a very long time. If we stop training DCGAN when the Inception score is maximal, we see no mode collapse, but the sample diversity is still lower than for OT-GAN (see figure 6).

---

### Official Review · AnonReviewer2 · 2017-11-27
**Design choices**

**Rating:** 6
**Confidence:** 2

**Review:**

The paper presents a variant of GANs in which the distance measure between the generator's distribution and data distribution is a combination of two recently proposed metrics. In particular, a regularized Sinkhorn loss over a mini-batch is combined with Cramer distance "between" mini-batches. The transport cost (used by the Sinkhorn) is learned in an adversarial fashion. Experimental results on CIFAR dataset supports the usefulness of the method.

The paper is well-written and experimental results are supportive (state-of-the-art ?)

A major practical concern with the proposed method is the size of mini-batch. In the experiment, the size is increased to 8000 instances for stable training. To what extent is this a problem with large models? The paper does not investigate the effect of small batch-size on the stability of the method. Could you please comment on this?

Another issue is the adversarial training of the transport cost. Could you please explain why this design choice cannot lead instability?

---

> ### Author Response · Authors · 2018-01-02
> **Happy new year and thank you for your review!**
>
> Regarding the two issues your raise:
>
> * In section 5.2 we present an experiment on CIFAR-10 where we vary the batch size used in training OT-GAN. As shown in Figure 4, small batch sizes are less stable during training, although the results are still on par with previous work on GANs. For large models we can reach the large batch sizes required for optimal results by using more GPUs and/or GPUs with more memory. Many of the other recent SOTA works in GANs use even more time and compute than we do, but it is indeed a limitation of the method as clearly indicated in the paper. In the updated paper we have expanded our discussion of this experiment, the causes behind its result, and its practical importance.
>
> * Our transport cost depends on a critic neural network v, which embeds the images (generated and real) into a latent space. As long as this embedding is one-to-one / non-degenerate, the statistical consistency guarantees associated with minimizing energy distance carry over to our setting. When the critic v is learned adversarially, the embedding could potentially degenerate and we could lose these properties. In practice, we manage to avoid this by updating the generator more often than the critic. If the critic maps two distinct inputs to similar embeddings, the generator will take advantage of this, thereby putting pressure on the critic to adapt the embedding. An alternative solution we tried is to parameterize the critic using a RevNet (Gomez et al. 2017): This way the mapping is always one-to-one by construction. Although this also works, we found it to be unnecessary when updating the generator often enough. The updated paper includes additional discussion on this point, and it also includes a new experiment (appendix C) further investigating the importance of adversarially learning the transport cost.

---

### Official Review · AnonReviewer3 · 2017-11-30
**An interesting paper on how to push optimal transport in the "classical" GAN framework**

**Rating:** 6
**Confidence:** 3

**Review:**

There have recently been a set of interesting papers on adapting optimal transport to GANs. This makes a lot of sense. The paper makes some very good connections to the state of the art and those competing approaches. The proposal makes sense from the generative standpoint and it is clear from the paper that the key contribution is the design of the transport cost. I have two main remarks and questions.

* Regarding the transport cost, the authors say that the Euclidean distance does not work well. Did they try to use normalised vectors with the squared Euclidean distance ? I am asking this question because solving the OT problem with cost defined as in c_eta is equivalent to using a *normalized squared* Euclidean distance in the feature space defined by v_eta. If the answer is yes and it did not work, then there is indeed a real contribution to using the DNN. Otherwise, the contribution has to be balanced. In either case, I would have been happy to see numbers for comparison.

* The square mini batch energy distance looks very much like a maximum mean discrepancy criterion (see the work of A. Gretton), up to the sign, and also to regularised approached to MMD optimisation (see the paper of Kim, NIPS'16 and references therein). The MMD is the solution of an optimisation problem which, I suppose, has lots of connections with the dual Wasserstein GAN. The authors should elaborate on the relationships, and eventually discuss regularisation in this context.

---

> ### Author Response · Authors · 2017-12-11
> **quick update**
>
> Thanks for your review! We're currently working on updating the paper, but I wanted to send you a quick reply regarding the two points you have raised:
>
> * We have now run exactly the experiment you suggest, using cosine distance in the pixel space (or equivalently squared Euclidean distance with normalized vectors) instead of in the critic-space defined by v_eta. The maximum inception score we were able to achieve using this setup on CIFAR-10 was 4.93 (compared to 8.47 achieved using the DNN critic v_eta). You can download the corresponding samples here: https://www.dropbox.com/s/e27uqj6ah7j9avq/sample_pixel_space.png?dl=1 We will include the results of this experiment in the upcoming update of the paper.
>
> * There is indeed a close connection between energy distance and MMD. When the energy distance is generalized to other distances between individual samples it becomes equivalent to MMD. This is explained in the following work, among other places: Sejdinovic, Dino, Bharath Sriperumbudur, Arthur Gretton, and Kenji Fukumizu. "Equivalence of distance-based and RKHS-based statistics in hypothesis testing." The Annals of Statistics (2013): 2263-2291.
> The novel part of our proposed minibatch energy distance is that it further generalizes the energy distance from individual samples to minibatches. This makes it conceptually different from the existing literature, including Kim, Been, Rajiv Khanna, and Oluwasanmi O. Koyejo. "Examples are not enough, learn to criticize! Criticism for interpretability." In Advances in Neural Information Processing Systems, pp. 2280-2288. 2016. (please let us know if you were referring to a different paper) Equivalently we can say our minibatch energy distance generalizes MMD from individual samples to minibatches, but we choose to take the energy distance perspective as it more closely connects to other work in this area (e.g. Cramer-GAN). We will include this discussion + the references in the upcoming update.

---

> > ### Author Response · Authors · 2018-01-02
> > **FYI, the promised changes to the paper have now been made.**

---

### Decision · Program_Chairs · 2018-01-29
**ICLR 2018 Conference Acceptance Decision**

**Decision:**

Accept (Poster)

**Comment:**

This is another paper, similar in spirit to the Wasserstein GAN and Cramer GAN, which uses ideas from optimal transport theory to define a more stable GAN architecture. It combines both a primal representation (with Sinkhorn loss) with a minibatch-based energy distance between distributions.
The experiments show that the OT-GAN produces sharper samples than a regular GAN on various datasets. While more could probably be done to distinguish the model from WGANs and Cramer GANs, this paper seems like a worthwhile contribution to the GAN literature and merits publication.